# MicroRNA Expression Prior to Biting in a Vector Mosquito Anticipates Physiological Processes Related to Energy Utilization, Reproduction and Immunity

**DOI:** 10.3390/insects14080700

**Published:** 2023-08-10

**Authors:** Sarah Marzec, Alden Siperstein, Angela Zhou, Christina M. Holzapfel, William E. Bradshaw, Megan E. Meuti, Peter A. Armbruster

**Affiliations:** 1Department of Biology, Georgetown University, Washington, DC 20057, USA; sm3679@georgetown.edu (S.M.); zz220@georgetown.edu (A.Z.); 2Department of Entomology, The Ohio State University, Columbus, OH 43210, USA; siperstein.1@osu.edu (A.S.); meuti.1@osu.edu (M.E.M.); 3Laboratory of Evolutionary Genetics, Institute of Ecology and Evolution, University of Oregon, Eugene, OR 97403, USA; mosquito@uoregon.edu (C.M.H.); bradshaw@uoregon.edu (W.E.B.)

**Keywords:** mosquito-borne disease, biting, anticipatory physiology, vector, *Culex pipiens*, blood feeding

## Abstract

**Simple Summary:**

Mosquitoes are able to transmit a wide variety of devastating pathogens when they bite and obtain blood from their vertebrate hosts. Therefore, identifying the physiological processes required for biting by vector mosquitoes can contribute to developing strategies to suppress biting behavior and prevent disease transmission. In this study, we investigate the differential expression of small regulatory RNAs (microRNAs) between different strains of the Northern house mosquito, *Culex pipiens*, which is a major vector of West Nile virus and filarial nematodes. We measured differential microRNA expression specifically in the context of a behavioral biting assay, using populations with previously documented differences in biting propensity and the ability to produce eggs without a blood meal. We identified eight differentially expressed microRNAs; six of these are implicated in regulating physiological processes related to energy utilization, reproduction, and immunity. Our results are strikingly similar to previous studies demonstrating increased expression of messenger RNA-encoding proteins involved in energy utilization in association with biting. Furthermore, while previous studies have identified changes in microRNA expression occurring after consuming a blood meal, ours is the first study to demonstrate anticipatory changes in microRNA expression before blood is consumed.

**Abstract:**

Understanding the molecular and physiological processes underlying biting behavior in vector mosquitoes has important implications for developing novel strategies to suppress disease transmission. Here, we conduct small-RNA sequencing and qRT-PCR to identify differentially expressed microRNAs (miRNAs) in the head tissues of two subspecies of *Culex pipiens* that differ in biting behavior and the ability to produce eggs without blood feeding. We identified eight differentially expressed miRNAs between biting *C. pipiens pipiens* (Pipiens) and non-biting *C. pipiens molestus* (Molestus); six of these miRNAs have validated functions or predicted targets related to energy utilization (miR8-5-p, miR-283, miR-2952-3p, miR-1891), reproduction (miR-1891), and immunity (miR-2934-3p, miR-92a, miR8-5-p). Although miRNAs regulating physiological processes associated with blood feeding have previously been shown to be differentially expressed *in response* to a blood meal, our results are the first to demonstrate differential miRNA expression *in anticipation* of a blood meal before blood is actually imbibed. We compare our current miRNA results to three previous studies of differential messenger RNA expression in the head tissues of mosquitoes. Taken together, the combined results consistently show that biting mosquitoes commit to specific physiological processes in anticipation of a blood meal, while non-biting mosquitoes mitigate these anticipatory costs.

## 1. Introduction

When female mosquitoes bite and then feed on the blood of their vertebrate hosts, they can transmit a wide range of human pathogens, including malaria parasites, many viruses, and filarial nematodes. The economic and public health burden resulting from these mosquito-transmitted pathogens remains staggering; it is estimated that 228 million individuals were infected with malaria parasites in Africa in 2020 [1], and approximately one-third of the Earth’s population is considered to be at risk for infection by the dengue virus [2]. The benefit of blood-feeding (biting) to the mosquito is clear: blood provides a rich nutritional resource that can be allocated to reproduction. At the same time, blood feeding also incurs substantial costs. These costs include energy allocated to host seeking [3] and biting [4], a heat shock response stimulated by ingesting hot blood [5,6], metabolic costs of excreting or sequestering the toxic heme and iron resulting from the breakdown of hemoglobin [7,8,9,10,11,12], and upregulating metabolic pathways in anticipation of an imminent blood meal, whether or not that blood is consumed [13,14]. The costs of blood feeding have likely promoted the evolution of a non-biting life history from blood-feeding ancestors, which has occurred multiple times in mosquitoes. For example, three genera of mosquitoes never bite (*Malaya*, *Topomyia*, and *Toxorhynchites*) and several facultatively or obligately non-biting species occur in genera comprised of mostly species that do bite [15,16,17,18,19,20,21,22,23]. Our long-term goal is to determine the molecular and physiological basis of the evolution of a non-biting life history in order to develop targeted genetic or biochemical interventions that reduce biting behavior in natural vector populations.

*C. pipiens pipiens* and *C. pipiens molestus* (hereafter, Pipiens and Molestus, respectively) are two sub-species of the vector mosquito *C. pipiens*; these sub-species differ in biting behavior and the ability to reproduce without biting [24,25,26,27]. Previously, we compared messenger RNA (mRNA) expression during a biting assay between biting Pipiens (actively probing) and non-biting Molestus (mosquitoes that had not landed on the host). Importantly, this previous study quantified differences in the expression of messenger RNA (mRNA) transcripts *before* blood was imbibed [14], thereby identifying *anticipatory* transcriptional responses of biting Pipiens relative to their non-biting Molestus counterparts. Of particular relevance to the current study, our previous study found that biting Pipiens exhibited an anticipatory transcriptional commitment to energy production by upregulating transcripts encoding proteins involved in oxidative phosphorylation, the citric acid (TCA) cycle, and multiple pathways feeding into the TCA cycle. These results are strikingly similar to those of de Carvalho et al. [4], who found that expression of mRNA transcripts involved in muscle development and oxidative phosphorylation, as well as the cellular processes of mitochondrial biogenesis and ATP-linked respiration, all increased in the head tissues of *A. aegypti* females between zero and four days post-eclosion as they approached competency to blood feed.

Here, we compare microRNA (miRNA) expression between biting Pipiens and non-biting Molestus, using the exact same RNA samples as in our previous mRNA study [14]. MicroRNAs are endogenously produced, non-coding RNAs approximately 18–24 nucleotides in length that post-transcriptionally regulate mRNAs. Most often, miRNAs cause degradation or translational repression of their mRNA targets, although in some cases, miRNAs can stabilize or even enhance the expression of mRNAs [28,29,30,31]. MicroRNAs thus provide a mechanism of post-transcriptional regulation of their target mRNAs before translation [32]. Furthermore, a single miRNA can regulate dozens or even hundreds of target mRNAs [33,34,35], indicating the capacity of miRNAs to modulate complex physiological processes [36]. Indeed, in the vector mosquito *A. aegypti*, several miRNAs are upregulated in response to blood feeding and have been demonstrated to target downstream physiological processes including blood digestion [37], nutritional provisioning of oocytes [38], and ovarian development [39].

In the current study, we perform small-RNA sequencing to examine differences in the miRNA expression between biting Pipiens and non-biting Molestus before blood is ingested, using the same biological samples as in the previous study of differences in mRNA expression described above [14]. Our results are distinct from previous results on the role of miRNAs in blood-feeding related processes because they reflect *anticipatory responses before* blood-feeding rather than downstream responses *after* blood has been ingested. Similar to our previous study of mRNAs [14], we identified several differentially expressed miRNAs that are implicated in energy utilization. Furthermore, in agreement with previous studies of mosquito miRNA expression in response to blood feeding, we also identified differentially expressed miRNAs implicated in physiological processes related to immunity and reproduction.

## 2. Materials and Methods

Insect colony maintenance and head tissue collections. The total RNA samples used in this study consisted of the exact same samples as those used in previously published work that analyzed the differences in mRNAs between biting Pipiens and non-biting Molestus [14]. In brief, the colonies of Molestus and Pipiens correspond to BG1 (autogenous *C. p. molestus*) and AG2 (anautogenous *C. p. pipiens*) from Noreuil and Fritz [24]. As described previously [14], our preliminary experiments confirmed that Pipiens females required a bloodmeal to complete ovary maturation. In contrast, >90% of Molestus females produced eggs without a blood meal within ~100 h of eclosion and did not attempt to bite a vertebrate host. We collected head tissues for RNA extraction from both non-biting females of Molestus and biting females of Pipiens three days post-adult-emergence during a one-hour period between Zeitgeber time (ZT) ZT12 and ZT15. To collect biting female Pipiens, any females that landed on a human blood source, probed their mouthparts into the source, and inserted their proboscis until the labium was bent and the fascicle was exposed, were aspirated into a collecting tube. It is important to note that these biting females were collected *before* they had the opportunity to imbibe any blood. To collect non-biting females of Molestus, we first discarded any females that attempted to bite a human blood source during the one-hour exposure period (0–5 females/cage; <1.7% of total females attempted to bite). We then aspirated the remaining non-biting Molestus into a collection tube. We collected three biological replicate samples of head tissues for both biting Pipiens and non-biting Molestus; each Pipiens replicate sample consisted of between 31 and 35 heads, and each Molestus replicate sample consisted of 35 heads.

RNA extraction, library preparation, and sequencing. Six frozen head tissue samples (three biting Pipiens and three non-biting Molestus) were homogenized in 500 μL of TRIzol (Invitrogen, Waltham, MA, USA). The samples were then shipped to the University of Oregon Genomics and Cell Characterization Core Facility (GC3F), where RNA extraction, library preparation, and sequencing were performed. RNA integrity was assessed on an RNA chip (Bioanalyzer 2100, Agilent Technologies, Santa Clara, CA, USA). Then, six small-RNA sequencing libraries were prepared with Perkin Elmer’s NextFlex small RNA-seq kit v3 (NOVA-5132-05) according to the manufacturer’s instructions. Size selection was performed according to the gel-free method as described in the manual. Finally, all six libraries were combined in equimolar ratios, and single-end, 160-bp reads were sequenced on a single lane of NovaSeq 6000. The raw reads are available under accession PRJNA883247 in NCBI’s short read archive (SRA).

Bioinformatics analyses. The bioinformatics workflow is described here [40]: https://github.com/srmarzec/Culex_Biting_miRNA/blob/main/MasterNotes.md (accessed on 25 April 2022).

Read Cleaning and Filtering. Briefly, reads were cleaned with Trimmomatic (version 0.39). Illumina small RNA NexFlex adapters were removed, and the following settings for single-end reads were applied; HEADCROP:4, TRAILING:10, SLIDINGWINDOW:4:15, MINLEN:17. The quality of the reads was confirmed using FastQC (v0.11.9), resulting in files consisting of between 30 and 60 million reads. Next, reads were filtered to remove tRNA and rRNA sequences by aligning the reads to the *Culex quinquefasciatus* complete mitochondrion genome (NCBI Reference Sequence: NC_014574.1) using bowtie2 (v2.4.4) and retaining all unaligned reads. The reads were then size-sorted with a custom Python script, retaining only reads that fell within the 18–24 base-pair size range, the expected size of mosquito miRNAs [41].

Identification of miRNAs with miRDeep2. Indexing for miRDeep2 was performed with Bowtie (v1.3.1) using the *C. quinquefasciatus* reference genome (GCF_015732765.1) obtained from NCBI. Known mature miRNA sequences were obtained for *C. quinquefasciatus* from miRBase and for *C. pipiens* and *C. quinquefasciatus* from Hong et al. [41]. Exact sequence duplicates were removed from the combined list, resulting in 144 unique known mature miRNA sequences (Appendix A). Based on these 144 reference miRNA sequences, mirRDeep2 generated a list of 122 unique precursor and mature miRNA pairs that the Pipiens and Molestus reads were then mapped against (miRDeep2.0.1.3 [42]; Appendix A).

Differential expression analysis. After obtaining read counts from miRDeep2, all downstream analysis was performed in R version 4.0.2. Read counts were transformed to a log2 scale using rlog, and then a principal components analysis (PCA) was performed using plotPCA (DESeq2 v1.30.1 [43]) to visualize the transcriptional profiles of Pipiens and Molestus. To identify differentially abundant miRNAs between Molestus and Pipiens, differential expression analysis of the 100 miRNAs that had at least 10 reads across all samples in Pipiens and Molestus was performed with DESeq2 (v1.30.1) [43]. Differentially expressed miRNAs were defined by a Benjamini-Hochberg false discovery rate adjusted *p*-value less than 0.05 and an absolute log2 fold change greater than 0.58 (i.e., absolute fold change = 1.5). Since Molestus was used as the control in the DESeq analysis, positive fold changes indicate the miRNA is more abundant in Pipiens.

Target Prediction. miRNA target sites are disproportionately located in the 3′ UTR region of mRNAs [44]. Therefore, a version of the *C. quinquefasciatus* reference genome (VectorBase-55_CquinquefasciatusJohannesburg) from VectorBase was used because it has annotated 3′ UTR regions. The consensus between two miRNA target site prediction algorithms was used as a conservative approach to identifying the putative regulatory targets of the differentially expressed miRNAs. First, miRanda [45] was used with the following parameters: score cutoff 140, energy cutoff −20, gap open penalty −9, gap extension penalty −4, scaling parameter 4. Second, RNAhybrid [46] was used with a calibration output from the RNACalibrate module and the following parameters: binding required in miRNA positions 2–7, *p*-value < 0.1, maximum target sequence length 100,000, energy cutoff −20.

Quantitative Real-Time PCR (qRT-PCR) validation. To validate the miRNA expression results obtained by RNAseq, we measured the expression of all eight miRNAs identified as differentially expressed (Table 1) in RNA samples from five independently collected biological replicates of biting Pipiens and non-biting Molestus (10 total samples). Each biological replicate sample used for qRT-PCR contained 6–13 female mosquito heads.

RNA was isolated using TRIzol reagent (Invitrogen) according to the manufacturer’s instructions. Complementary DNA (cDNA) was synthesized with the miRCURY LNA RT Kit (Qiagen, Germantown, MD, USA) using 20 ng of RNA and the 5X Sybr Green Reaction Buffer following the manufacturer’s protocol. The relative expression of miRNAs was measured using miRCURY LNA miRNA PCR assays (Qiagen). Custom primers were designed to measure the relative expression of novel miR-1 and novel miR-4. For the remaining miRNAs, we used existing miRNA primers that were designed for other mosquitoes that exactly matched the mature miRNA sequences in *C. pipiens.* All primer sequences are provided in Appendix A. The relative expression of each target miRNA was normalized to the expression of let-7, as in earlier studies of miRNA expression in *C. pipiens* [51]. All qRT-PCR reactions were run in triplicate, containing 5 μL of 2X miRCURY SybrGreen Master Mix, 1 μL of primer, 1 μL of water, and 1 μL of diluted cDNA (10 μL total volume). The samples were run on a CFX Connect qRT-PCR machine (BioRad, Hercules, CA, USA) using the following profile; a 2-min denaturation at 95 °C, followed by 40–50 cycles of 95 °C for 10 s and 56 °C for 60 s. Following qRT-PCR amplification, a melt curve was run to confirm that only a single product was produced.

The qRT-PCR data were analyzed as previously described [14,51]. In brief, after calculating the mean and standard deviation cycle quantification level (Cq) among three technical replicates, the relative expression of each miRNA within a biological replicate was normalized to the expression of let-7 for the same biological replicate (2^−ΔCT^ method [52]). The average relative miRNA expression and standard error among biting Pipiens and non-biting Molestus were calculated, and the relative fold change was obtained by dividing the average relative miRNA expression in biting Pipiens samples by the average relative expression in non-biting Molestus samples. Student’s *t*-tests were used to calculate significant differences between the relative expression of miRNAs in biting Pipiens and non-biting Molestus.

## 3. Results

### 3.1. Small RNA Sequencing

Small RNA sequencing of three biological replicate samples of non-biting Molestus head tissue and three biological replicate samples of biting Pipiens head tissue produced between 48,470,972 and 89,622,867 reads per biological replicate. Between 79.15–85.01% of these reads per biological replicate were retained after filtering with Trimmomatic. Between 0.32% and 0.49% of the reads were removed as tRNA or rRNA contaminants. After size filtering, a total of 15,546,853–36,760,493 reads per sample were retained (Appendix A). One hundred miRNAs had at least ten reads mapped by miRDeep2 across the six samples and were used in differential expression analysis (Appendix A).

Principal component analysis showed that the miRNA expression profiles of biting Pipiens versus non-biting Molestus samples clustered separately on the first principal component axis, which accounted for 58% of the variance (Appendix A). Overall, eight miRNAs were significantly differentially expressed between biting Pipiens and non-biting Molestus (Figure 1; adjusted *p*-value < 0.05, absolute fold change > 1.5). Out of these eight, it was possible to obtain functional annotations for seven miRNAs (Table 1).

### 3.2. MicroRNA Target Prediction

A total of 2783 3′ UTRs were identified from the reference genome annotation and used with the target prediction software. For the eight differentially expressed miRNAs in Table 1, miRanda predicted 57 target genes, and RNAhybrid predicted 375 gene targets. Thirty-two genes were present in both sets of predicted targets between these two software programs (Appendix A).

### 3.3. Quantitative Reverse-Transcription PCR

The qRT-PCR analyses confirmed the miRNA short-read sequencing results; seven of eight miRNAs were differentially abundant in the same direction and displayed similar relative fold changes as the RNAseq results (Appendix A). The one exception is miR-2952-3p, which was detected in four out of five of the non-biting Molestus biological replicates by qRT-PCR but in none of the biting Pipiens biological replicates, even after 50 rounds of qRT-PCR amplification. This result is consistent with the RNAseq findings, where miR-2952-3p was more abundant in Molestus relative to Pipiens.

## 4. Discussion

A previous study examining mRNA differences in head tissues between biting Pipiens and non-biting Molestus showed anticipatory upregulation of energy-production pathways in Pipiens and upregulation of fructose and mannose metabolism in Molestus [14]. In the current study, we identified eight miRNAs that were differentially expressed between Pipiens and Molestus based on RNAseq (Table 1) and qRT-PCR (Appendix A). Two of the eight are novel miRNAs with targets that are either uncharacterized (novel-miR4) or have an ambiguous functional significance relative to blood feeding (novel-miR1). Five of the eight are less abundant in biting Pipiens than non-biting Molestus. These five differentially expressed miRNAs are predicted to promote processes related to energy utilization and immunity in biting Pipiens and suppress these processes in non-biting Molestus. One differentially expressed miRNA is more abundant in biting Pipiens and is implicated in promoting fecundity and longevity based on experimental evidence from another vector mosquito, *Aedes albopictus* (Table 1, Figure 2).

In addition to biting behavior and reproductive physiology, Pipiens and Molestus also differ in a suite of ecophysiological traits, including mating behavior and above- vs. below-ground habitat utilization [25,26]. Therefore, expression differences detected in our previous mRNA study and in the current comparison between biting Pipiens and non-biting Molestus could potentially be due to factors other than biting behavior and reproductive physiology [27]. However, this interpretation is unlikely for several reasons. First, our samples were collected specifically in the context of a behavioral biting assay using populations with previously documented differences in biting propensity and the ability to produce eggs without a blood meal [24]. Second, in our previous study, we showed that mRNA expression differences between biting Pipiens and non-biting Molestus strongly overlapped with mRNA expression differences between biting and non-biting populations of the mosquito *Wyeoymia smithii*. Moreover, our previous mRNA results [14], current microRNA (miRNA) results in *C. pipiens*, and the results of de Carvalho et al. [4] in *A. aegypti* all identify energy utilization as an important process associated with biting in head tissues. Taken together, these overlapping results in three evolutionarily diverse mosquito genera (*Culex*, *Aedes, Wyoemyia*) strongly support a conserved molecular and physiological response to differences in biting behavior rather than simply specific differences between Pipiens and Molestus. Finally, the miRNA differences we detected in the current study between biting Pipiens and non-biting Molestus show clear functional relevance of energy metabolism and immunity to an anticipated blood meal.

Blood feeding is expected to be energetically expensive, in part due to the activities of muscles in the mosquito head. These activities include coordinating movements of the sensory organs and mouthparts during the exploratory phase after the mosquito has landed on a vertebrate host, as well as penetration of the fascicle into the host epidermis and activation of the cibarial and pharyngeal pumps for sucking blood [53]. Our previous mRNA results described above are consistent with these energetic requirements [14], as are the similar results of de Carvalho et al. [4] in *A. aegypti*. In de Carvalho et al.’s [4] study, females of *A. aegypti* from zero to four days post-eclosion approaching competency to blood feed exhibited upregulation of mRNA transcripts involved in both muscle development and oxidative phosphorylation. Also during this period, de Carvalho et al. [4] found dramatic mitochondrial biogenesis and increased ATP-linked respiration in head tissues. Our current results (described below) in *C. pipiens* demonstrate that energy utilization associated with blood feeding is regulated at the miRNA level in addition to the mRNA level. This discovery warrants further investigation into *A. aegypti* and other vector mosquitoes.

The three miRNAs implicated in promoting energy utilization in Pipiens and repressing energy utilization in Molestus are miR-283, miR-8-5p, and miR-2952-3p (Table 1). All three of these miRNAs were less abundant in Pipiens relative to Molestus, indicating that translation of the targets of these miRNAs is expected to be released from suppression in biting Pipiens. This released suppression would lead to enhanced translation of transcripts that enhance energy utilization in biters (Figure 2). The first of these miRNAs, miR-283, is predicted to target ATP synthase lipid-binding protein transcripts (Appendix A). Consistent with this prediction, our previous mRNA study showed that three ATP synthase transcripts were significantly upregulated in biting Pipiens [14]. The second miRNA, miR-8-5p, is predicted to target the adipokinetic hormone pathway (AKH, Appendix A). AKH functions similarly to glucagon in mammals by mobilizing lipid and carbohydrate energy reserves (reviewed by [54]). Furthermore, Nouzova et al. [49] found binding sites of miR-8-5p within the AKH-1 transcripts of *A. aegypti*. Finally, miR-2952-3p is predicted to target hydroxyacylglutathione hydrolase transcripts (Appendix A). This enzyme is a critical component of pyruvate metabolism, where it converts methylglyoxal to lactic acid and reduced glutathione [55]. Furthermore, in our previous mRNA study, eight of 11 differentially expressed genes in the pyruvate pathway were upregulated in biting Pipiens [14]. Taken together, these results corroborate our previous results from mRNA expression in head tissues [14], as well as those of de Carvalho et al. [4] in *A. aegypti*, indicating increased energy utilization in the head tissue of biting Pipiens relative to non-biting Molestus.

In contrast to the results discussed above, a single miRNA potentially related to energy utilization, miR-1891, was upregulated in Pipiens relative to Molestus. The predicted target of miR-1891 is trehalose transporter1 (Tret1) (Appendix A). Consistent with the differential expression of miR-1891, our previous analyses of mRNAs showed that the expression of the Tret1 transcript was lower in biting Pipiens than in non-biting Molestus [14]. These results imply increased Tret1 protein levels in non-biting Molestus relative to biting Pipiens. Taken together with our previous mRNA results showing upregulation of fructose and mannose metabolism in non-biting Molestus [14] and the established role of sugar feeding in supporting autogenous reproduction [16], these results imply that non-biting Molestus prioritize sugar metabolism to generate sufficient energy reserves to reproduce without a blood meal.

In addition to its role in regulating trehalose transport, levels of miR-1891 affect female reproduction in the mosquito *A. albopictus*. Puthiyakunnon et al. [48] found that knockdown of miR-1891 in adult females caused an approximately two-fold shorter adult lifespan and a four-fold decline in initial fecundity relative to control females, showing that high levels of miR-1891 are necessary for achieving maximal reproductive success. Our results showing increased expression of miR-1891 in biting Pipiens relative to non-biting Molestus (Table 1) indicate a conserved function in *C. pipiens* and clarify that upregulation of this miRNA occurs before the blood meal is imbibed rather than as a response to consuming blood itself.

Feeding on vertebrate blood represents a significant immune challenge for mosquitoes [56,57]. We identified three miRNAs with immunity-related functions that were less abundant in the head tissues of biting Pipiens relative to non-biting Molestus (Table 1, Figure 2). Two of these miRNAs appear to promote immunity-related processes in biters and block immunity-related processes in non-biters. Although transcripts encoding immune peptides are usually expressed in the fat body or hemocytes, it has become clear that a wide variety of tissue types are capable of producing immune transcripts that could be targeted by the miRNAs we identified [58]. Thus, the immunity-related miRNAs we identified could be expressed in hemocytes circulating in the hemolymph of the head or in other head tissues.

The first immunity-related miRNA we found to be less abundant in biting Pipiens relative to non-biting Molestus is miR-2941-3p (Table 1). Thus, translation of miR-2941-3p targets is expected to be released from suppression in biting Pipiens vs. non-biting Molestus, leading to enhanced translation of transcripts associated with immune responses in biters (Figure 2). Similar to our results, Liu et al. [59] showed that miR-2941-3p was downregulated in the midgut of the Asian tiger mosquito, *A. albopictus*, that was infected with dengue virus. Furthermore, our analyses predicted that miR-2941-3p targets defensin-C in *C. pipiens* (LOC6032313; Appendix A). Defensin proteins are primarily involved in innate immune responses within the midguts of mosquitoes, including *A. aegypti* [60,61]. As described above, miR-8-5p is also less abundant in biting Pipiens relative to non-biting Molestus. In addition to the previously discussed role of miR-8-5p in regulating energy utilization, differential expression of this miRNA is consistent with an increased immune response associated with biting (Figure 2). Previously, Monsanto-Hearne et al. [50] demonstrated that miR-8-5p is downregulated in response to viral infection, and decreased expression of miR-8-5p increases the level of the Drosophila Jun mRNA transcripts. The dJun protein is a transcription factor that is involved in the innate immune response, activating the JNK signaling pathway, and thereby upregulating stress responses [62,63]. More recently, Soory and Ratnaparkhi [64] have shown that Jun also increases the transcription of antimicrobial peptides. Our findings show that biting Pipiens preemptively suppresses miR-2941-3p and miR-8-5p, likely leading to an increase in the expression and/or translation of defensin and Jun transcripts, indicating that these females are preparing to upregulate immune responses prior to imbibing vertebrate blood.

miR-92a is also less abundant in biting Pipiens relative to non-biting Molestus (Table 1). In addition to suppressing the translation of endogenous mRNA transcripts, miRNAs can directly bind to viral genomes and inhibit viral replication [65]. Furthermore, Yen et al. [47] used computational analyses to predict that miR-92a in *A. aegypti* binds to the positive-sense genomic viral RNA encoding the non-structural protein 5 (NS-5) of dengue virus I. Although *C. pipiens* does not transmit dengue, it is an important vector of other positive-sense RNA arboviruses that cause West Nile fever, Eastern equine encephalitis, and St. Louis encephalitis [66,67,68,69]. If miR-92a does indeed inhibit RNA viral replication in *C. pipiens*, it is not clear why it would be less abundant in biting Pipiens relative to non-biting Molestus. One likely possibility is that miR-92a is unregulated in biting Pipiens after a blood meal is actually imbibed and/or in alternative tissues such as the midgut.

## 5. Conclusions

We identified eight miRNAs that differed in expression between the head tissues of biting Pipiens and non-biting Molestus. Of these eight miRNAs, six have validated functions or predicted mRNA targets related to energy utilization, reproduction, and immunity. Differences in the expression of miRNAs promoting energy utilization in the head tissues of biting Pipiens relative to non-biting Molestus are consistent with previous studies at the mRNA level of these same tissues [14], as well as studies of the head tissues of *A. aegypti* before attaining competency for blood feeding [4]. Taken together, these results emphasize the dramatic energetic investment that is required for blood-feeding mosquitoes to locate a blood-meal host and acquire blood. Previous studies have identified miRNAs regulating energy utilization, immunity, and reproductive processes in mosquitoes *after* a blood meal has been consumed [37,38,39,48,70], but our results are the first to demonstrate that *anticipatory* miRNA-mediated regulation of these processes occurs *before* blood consumption has even commenced. An important future goal is to determine whether these anticipatory responses at the miRNA level occur in other vector species, at other developmental time points, and in other tissues (i.e., midgut, fat body, and ovaries).

## Figures and Tables

**Figure 1 insects-14-00700-f001:**
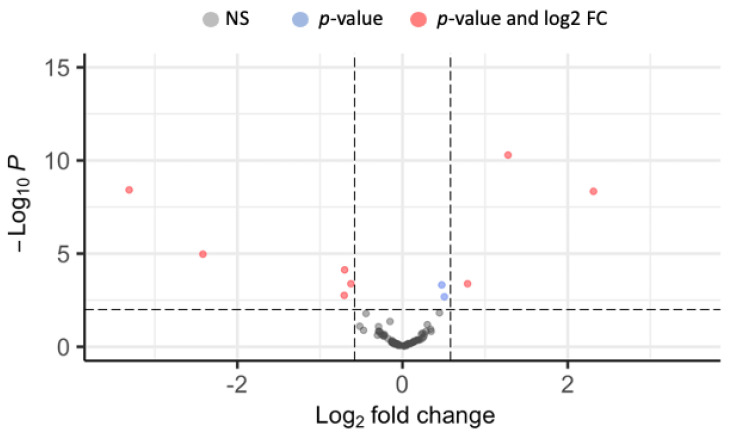
Differential expression of 100 miRNAs in biting Pipiens and non-biting Molestus. Each point represents a single miRNA. Gray points indicate no significant differences in expression. Blue points represent miRNAs with statistically significant expression differences, but low absolute log_2_ fold-change values. Red points indicate the eight statistically significant differentially expressed miRNAs with an absolute fold change value greater than 1.5. Positive values indicate greater expression levels in biting Pipiens relative to non-biting Molestus.

**Figure 2 insects-14-00700-f002:**
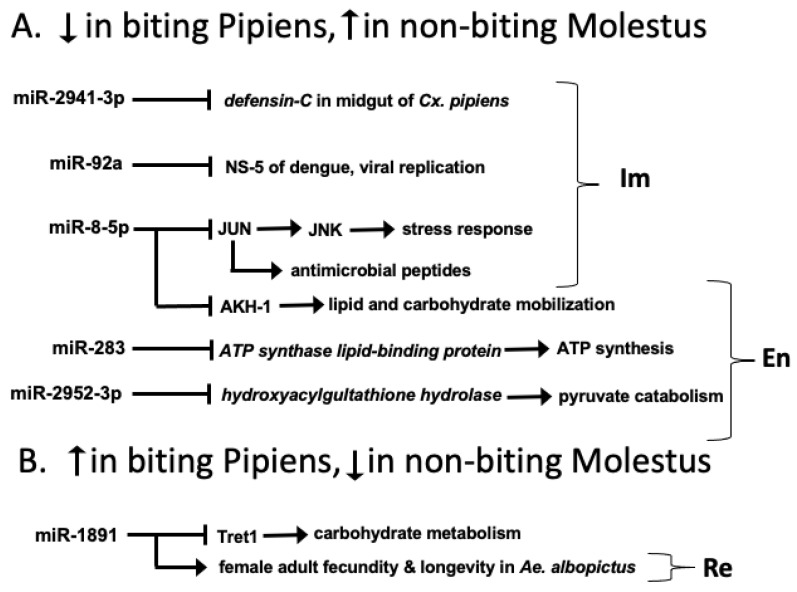
Differential miRNA expression in biting Pipiens vs. non-biting Molestus. (**A**) microRNAs downregulated (↓) in biting Pipiens and upregulated (↑) in non-biting Molestus; (**B**) microRNAs up-regulated (↑) in biting Pipiens and downregulated (↓ ) in non-biting Molestus. Im, immune function; En, energy metabolism; Re, reproduction. functions to inhibit; ⟶ functions to stimulate. For example, lower expression of miR-2941-3p in biters relative to nonbiters implies released suppression of defensin-C in biters. See text for further details.

**Table 1 insects-14-00700-t001:** Biological function of significant differentially expressed miRNAs. Predicted and/or validated targets and functions for seven of eight significant miRNAs with their respective fold change for Pipiens relative to Molestus. miRNAs are grouped into functional categories based on biological function. Biological function is in mosquitoes unless otherwise specified. Predicted targets identified in this study are based on two independent analyses. Positive values indicate increased miRNA expression in biting Pipiens relative to non-biting Molestus. Adjusted *p*-value < * 0.01; ** 1.0 × 10^−3^; *** 1.0 × 10^−8^.

miRNA	Fold Change	Putative Biological Function	Functional Category
miR-92a	−1.542 **	Predicted Target: NS5 region of DENV-1 [47]	Immunity
miR-1891	1.725 **	Validated function: Fecundity and adult longevity [48], Predicted Target: Facilitated trehalose transporter Tret1	Reproduction Energy Utilization
miR-2941-3p	−5.341 **	Predicted Target: defensin-C	Immunity
miR-8-5p	−1.625 **	Predicted Target: AKH1 [49],	Energy Utilization
		Validated Target: Drosophila Jun, dJun [50] in Drosophila	Immunity
miR-283	−1.631 *	Predicted Target: ATP synthase lipid-binding protein	Energy Utilization
miR-2952-3p	−9.911 ***	Predicted Target: Hydroxyacylglutathione hydrolase	Energy Utilization
novel-miR1	2.425 ***	Predicted Target: ankyrin repeat and MYND domain-containing protein 2	-
novel-miR4	4.962 ***	Predicted Target: CPIJ003731, uncharacterized protein	-

## Data Availability

The raw reads are available in NCBI’s short read archive (SRA) under accession PRJNA883247.

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
