# Peer review of "MicroRNA Expression Prior to Biting in a Vector Mosquito Anticipates Physiological Processes Related to Energy Utilization, Reproduction and Immunity"

_insects, 2023, doi:10.3390/insects14080700_

Round 1

Reviewer 1 Report

This is a well written and executed manuscript on an interesting topic. Your manuscript will be a useful guide to people who will be doing future microRNA research in mosquitoes. I have a few minor comments, both here and on the MS itself.

Abstract

Please clarify the Pipiens/Molestus differences as there may be some readers who may not know about the differences in behaviour of these two subspecies.

Methods.

Please give a bit more details about the quality control used for the RNA extraction please.

Discussion

There should not be figures in the discussion. Please move this as a summary into the results.

Author Response

Note: review comments in black font, author responses in red font

This is a well written and executed manuscript on an interesting topic. Your manuscript will be a useful guide to people who will be doing future microRNA research in mosquitoes. I have a few minor comments, both here and on the MS itself.

Thank you!

(I) Abstract: Please clarify the Pipiens/Molestus differences as there may be some readers who may not know about the differences in behaviour of these two subspecies.

We have modified the text on lines 29-32 of the abstract to read as follows: “Here, we conduct small-RNA sequencing and qRT-PCR to identify differentially expressed microRNAs (miRNAs) in head tissues of two subspecies of Culex pipiens that differ in biting behavior and the ability to produce eggs without blood feeding. We identified eight differentially expressed miRNAs between biting Cx. pipiens pipiens (Pipiens) and non-biting individuals of Cx. pipiens molestus (Molestus); six of these miRNAs have validated functions or predicted targets related to energy utilization (miR8-5-p, miR-283, miR-2952-3p, miR-1891), reproduction (miR-1891), and immunity (miR-2934-3p, miR-92a, miR8-5-p).”

(II) Methods: Please give a bit more details about the quality control used for the RNA extraction please.

We have added the following text to lines 137-138 of the methods section: “First, RNA integrity was assessed on an RNA chip (Bioanalyzer 2100, Agilent Technologies, Santa Clara California).”

(III) Discussion:There should not be figures in the discussion. Please move this as a summary into the results.

We have moved Figure 2 to the end of the results section, before the discussion.

Reviewer 2 Report

The work by Marzec et al. is a logical extension of their previous work on mRNA expression in biting and non-biting C. pipiens. Here, they show in a 1h period at 3d following emergence that there are 8 differentially expressed miRNAs in mosquito heads (3 biological reps). The replication is good and the data analysis is solid. However, I have some concerns that need to be addressed to improve the impact and support the interpretations presented. The authors note that the differential miRNAs are "anticipatory," but also suggest that miRNA expression might change with ingestion of the bloodmeal. If there is a change (on vs off, off vs on relative to the timepoint tested), it would seem that this miRNA expression does not "anticipate" feeding or correspond in the ways suggested to mRNA expression. This question should be resolved with a timecourse of at least two additional timepoints, including one preceding the one examined and one that extends into bloodmeal ingestion. The authors looked only at head miRNAs, but suggest that energy metabolism, immunity, etc. are the targets of these miRNAs. Certainly, there is communication between mosquito tissues, but at least one additional tissue (the midgut for example given this is the entry point for blood processing) should be included with these data to support the authors' interpretations. Without a timecourse and at least one other tissue, the discussion is uncomfortably speculative. Finally, only the differential miRNAs are presented and discussed in the main text. Were there miRNAs that were similarly expressed in both sub-species that were surprising? It would be good to include this context, too, because the metabolic targets of some of these miRNAs sit in pathways that are regulated homeostatically. Accordingly, if one protein in the pathway is regulated, would the overall effect be substantial or modest? Hard to say, but context would be helpful. Again, multiple timepoints and at least one other tissue should be included to improve the biological interpretation and more strongly suggest testable hypotheses. The writing is excellent, and the work is well-presented. The findings are exciting as preliminary, but publication without additional results is premature based on the arguments above. 

Author Response

Note: reviewer comments in black font, author's responses red font.

The work by Marzec et al. is a logical extension of their previous work on mRNA expression in biting and non-biting C. pipiens. Here, they show in a 1h period at 3d following emergence that there are 8 differentially expressed miRNAs in mosquito heads (3 biological reps). The replication is good and the data analysis is solid. However, I have some concerns that need to be addressed to improve the impact and support the interpretations presented.

Thank you for your feedback.

The authors note that the differential miRNAs are "anticipatory," but also suggest that miRNA expression might change with ingestion of the bloodmeal. If there is a change (on vs off, off vs on relative to the timepoint tested), it would seem that this miRNA expression does not "anticipate" feeding or correspond in the ways suggested to mRNA expression. This question should be resolved with a timecourse of at least two additional timepoints, including one preceding the one examined and one that extends into bloodmeal ingestion.   

We agree that it would be ideal to measure miRNA expression at multiple time points.  However, to do so is not currently feasible, and would cause a substantial delay in sharing our results with the research community.  As noted in multiple locations in the manuscript, the novelty of the current study is that it is the first to measure miRNA expression related to blood feeding before blood is ingested.  We hope that our study will stimulate additional work on this topic, including more fine-scale temporal sampling.  In response to this comment, and in order to emphasize the importance of fine-scale temporal sampling of miRNA expression in future studies, we have modified the final sentence of the manuscript to read as follows (lines 435-438): “An important future goal is to determine whether these anticipatory responses at the miRNA level occur in other vector species, at other developmental time points, and in other tissues (i.e., midgut, fat body, and ovaries).”

The authors looked only at head miRNAs, but suggest that energy metabolism, immunity, etc. are the targets of these miRNAs. Certainly, there is communication between mosquito tissues, but at least one additional tissue (the midgut for example given this is the entry point for blood processing) should be included with these data to support the authors' interpretations. Without a timecourse and at least one other tissue, the discussion is uncomfortably speculative.

Our interpretation of the significance of the expression of miRNAs related to energy utilization in head tissues is corroborated by evidence from two other studies: 1) Siperstein et al. 2022 (at the mRNA level, using the same tissue samples as used in this study), and 2) de Carvalho et al.,  2021 (at the mRNA level, in head tissues of Ae. aegypti).

One lines 319-324, we explain why blood feeding is expected to be energetically expensive in head tissue: “Blood feeding is expected to be energetically expensive in part due to the activities of muscles in the mosquito head. These activities include coordinating movements of the sensory organs and mouthparts during the exploratory phase after the mosquito has landed on a vertebrate host, as well as penetration of the fascicle into the host epidermis, and activation of the cibarial and pharyngeal pumps for sucking blood [49].”

On lines 352-355, we emphasize that our results are corrobarted by the two previous studies referred to above: “Taken together these results corroborate our previous results from mRNA expression in head tissues [14], as well as those of de Carvalho et al. [4] in Ae. aegypti, indicating increased energy utilization in the head tissue of biting Pipiens relative to non-biting Molestus.”

This issue is also addressed on lines 80-84 of the introduction, referning to our previous mRNA results in Cx. pipiens: “These results are strikingly similar to those of de Carvalho et al. [4], who found that expression of mRNA transcripts involved in muscle development and oxidative phosphorylation, as well as the cellular processes of mitochondrial biogenesis and ATP-linked respiration, all increased in head tissues of Ae. aegypti females between zero to four days post-eclosion as they approached competency to blood feed.” 

Our interpretation of the upregulation of miRNAs related to immunity in head tissue is addressed on lines 380-385: “Although transcripts encoding immune peptides are usually expressed in the fat body or hemocytes, it has become clear that a wide variety of tissue types are capable of producing immune transcripts that could be targeted by the miRNAs we identified [56]. Thus, the immunity-related miRNAs we identified could be expressed in hemocytes circulating in the hemolymph of the head, or in other head tissues.”

Finally, we agree with the importance of investigating upregulation of miRNAs in advance of a blood meal in other tissues, as emphasized in the final sentence of the manuscript (lines 435-438): “An important future goal is to determine whether these anticipatory responses at the miRNA level occur in other vector species, at other developmental time points, and in other tissues (i.e., midgut, fat body, and ovaries).”

Finally, only the differential miRNAs are presented and discussed in the main text. Were there miRNAs that were similarly expressed in both sub-species that were surprising? It would be good to include this context, too, because the metabolic targets of some of these miRNAs sit in pathways that are regulated homeostatically. Accordingly, if one protein in the pathway is regulated, would the overall effect be substantial or modest? Hard to say, but context would be helpful.  

We appreciate this suggestion, but feel that discussion of the biological significance of miRNAs that were expected to be differentially expressed but were not would be overly speculative, would make the manuscript too long, and would detract from emphasizing the significance of the miRNAs that were differentially expressed.  

Again, multiple timepoints and at least one other tissue should be included to improve the biological interpretation and more strongly suggest testable hypotheses. The writing is excellent, and the work is well-presented. The findings are exciting as preliminary, but publication without additional results is premature based on the arguments above. 

Thank you for your feedback.  As indicated above, it is not currently feasible to perform additional studies using multiple time points and tissues types. As noted above, we emphasize that the novelty of this study is that it is the first to identify miRNA expression related to blood feeding that occurs in advance of a blood meal. We hope that our results will stimulate future work on this topic, including studies examining miRNA expression at multiple time points and in multiple tissue types in advance of a blood meal.

Reviewer 3 Report

The manuscript by Marzec et al. entitled “MicroRNA expression prior to biting in a vector mosquito anticipates physiological processes related to energy utilization, reproduction and immunity” analyzes the differential expression of miRNAs in two subspecies of Culex pipiens (Pipiens and Molestus) whose females have different nutritional requirements to reproduce. While Pipiens females must feed on vertebrate blood to reproduce, the Molestus females may produce eggs without a blood meal. The experimental design allows the authors to demonstrate the anticipatory character in the changes of microRNA expression before the blood is consumed.

Comments to the authors:

The paper is written in a very good English and is easy to follow. 

The methodology and analyses are robust and include both biological and technical replicates.

I only have two comments:

1.     I have found the third paragraph of the Introduction to be more appropriate for the discussion section since they refer to results that have not yet been presented: “Therefore, expression differences detected in our previous mRNA study and in the current comparison between biting Pipiens and non-biting Molestus could potentially be due to factors other than biting behavior and reproductive physiology [27]. However, this interpretation is unlikely for several reasons. First, our samples were collected specifically in the context of a behavioral biting assay using populations with previously documented differences in biting propensity and the ability to produce eggs without a blood meal [24]. [..] Finally, the miRNA differences we detected in the current study between biting Pipiens and non-biting Molestus show clear functional relevance of energy metabolism and immunity to an anticipated blood meal.” 

2.     I have found some discussion missing about the functionality of the two novel miRNAs. Although there is no previous literature, I think authors have predicted a possible target for each one of them (as can be seen in the Supplemetary Datafile).

Minor comments and typos:

Please, review the list of author names carefully. In the list of authors (L.6) Peter Armbruster appears, but in the author contributions (L.441-L.444) he is listed as PAA (with an additional A). In addition, in the supplementary material, the same author is named as Peter A. Armbruster. 

References 22, 52, 61, 68 and 69. The titles of these references must be typed in lower case to be consistent with the rest of references.

Figure S.1 Change “Principle” to “Principal

Author Response

Note: reviewer comments in black font, author's response in red font.

The manuscript by Marzec et al. entitled “MicroRNA expression prior to biting in a vector mosquito anticipates physiological processes related to energy utilization, reproduction and immunity” analyzes the differential expression of miRNAs in two subspecies of Culex pipiens (Pipiens and Molestus) whose females have different nutritional requirements to reproduce. While Pipiens females must feed on vertebrate blood to reproduce, the Molestus females may produce eggs without a blood meal. The experimental design allows the authors to demonstrate the anticipatory character in the changes of microRNA expression before the blood is consumed.

Comments to the authors:

The paper is written in a very good English and is easy to follow. 

The methodology and analyses are robust and include both biological and technical replicates.

I only have two comments:

  1. I have found the third paragraph of the Introduction to be more appropriate for the discussion section since they refer to results that have not yet been presented: “Therefore, expression differences detected in our previous mRNA study and in the current comparisonbetween biting Pipiens and non-biting Molestus could potentially be due to factors other than biting behavior and reproductive physiology [27]. However, this interpretation is unlikely for several reasons. First, our samples were collected specifically in the context of a behavioral biting assay using populations with previously documented differences in biting propensity and the ability to produce eggs without a blood meal [24]. [..] Finally, the miRNA differences we detected in the current study between biting Pipiens and non-biting Molestus show clear functional relevance of energy metabolism and immunity to an anticipated blood meal.” 

As suggested, we have moved the paragraph referred to above to the Discussion section (Lines 298-318).

  1. I have found some discussion missing about the functionality of the two novel miRNAs. Although there is no previous literature, I think authors have predicted a possible target for each one of them (as can be seen in the Supplemetary Datafile).

 Thank you for point this out.  We have added the targets of novel-miR1 and novel-miR4 to Table 1. We have also added the following sentence to the discussion (lines  289-291): “Two of the eight are novel miRNAs with targets that are either uncharacterized (novel-miR4) or have an ambiguous functional significance relative to blood feeding (novel-miR1).”

Minor comments and typos:

Please, review the list of author names carefully. In the list of authors (L.6) Peter Armbruster appears, but in the author contributions (L.441-L.444) he is listed as PAA (with an additional A). In addition, in the supplementary material, the same author is named as Peter A. Armbruster. 

Thank you for pointing this out.  All references to Armbruster are now consistent (Peter A. Armbruster, PAA)

References 22, 52, 61, 68 and 69. The titles of these references must be typed in lower case to be consistent with the rest of references.

The titles of all references have been typed in lower case 

Round 2

Reviewer 2 Report

Best wishes with the work.